# Discovering Rare Genes Contributing to Cancer Stemness and Invasive Potential by GBM Single-Cell Transcriptional Analysis

**DOI:** 10.3390/cancers11122025

**Published:** 2019-12-16

**Authors:** Lin Pang, Jing Hu, Feng Li, Huating Yuan, Min Yan, Gaoming Liao, Liwen Xu, Bo Pang, Yanyan Ping, Yun Xiao, Xia Li

**Affiliations:** 1College of Bioinformatics Science and Technology, Harbin Medical University, Harbin 150081, China; panglin@hrbmu.edu.cn (L.P.); hjistb@gmail.com (J.H.); feng_li1988@hotmail.com (F.L.); YuanHT_tien@163.com (H.Y.); 15774512921@163.com (M.Y.); lgm179496478@163.com (G.L.); liberty_crystal@163.com (L.X.); pangbo@ems.hrbmu.edu.cn (B.P.); pingyanyan850904@163.com (Y.P.); 2Key Laboratory of Cardiovascular Medicine Research, Harbin Medical University, Ministry of Education, Harbin 150086, China

**Keywords:** single-cell RNA sequencing, rare genes, glioblastoma (GBM), cancer stemness, invasion

## Abstract

Single-cell RNA sequencing presents the sophisticated delineation of cell transcriptomes in many cancer types and highlights the tumor heterogeneity at higher resolution, which provides a new chance to explore the molecular mechanism in a minority of cells. In this study, we utilized publicly available single-cell RNA-seq data to discover and comprehensively dissect rare genes existing in few glioblastoma (GBM) cells. Moreover, we designed a framework to systematically identify 51 rare protein-coding genes (PCGs) and 47 rare long non-coding RNAs (lncRNAs) in GBM. Patients with high expression levels of rare genes like *CYB5R2* and *TPPP3* had worse overall survival and disease-free survival, implying their potential implication in GBM progression and prognosis. We found that these rare genes tended to be specifically expressed in GBM cancer stem cells, which emphasized their ability to characterize stem-like cancer cells and implied their contribution to GBM growth. Furthermore, rare genes were enriched in a 17-cell subset, which was located in an individual branch of the pseudotime trajectory of cancer progression and exhibited high cell cycle activity and invasive potential. Our study captures the rare genes highly expressed in few cells, deepens our understanding of special states during GBM tumorigenesis and progression such as cancer stemness and invasion, and proposes potential targets for cancer therapy.

## 1. Introduction

Glioblastoma (GBM) is the most common primary brain cancer in adults and the leading cause of brain cancer-related deaths, with median overall survival of only 12~18 months [1]. Increasing evidence suggests that GBM is more complicated than previously thought, being comprised of morphologically and phenotypically diverse cells [2,3] and rendering the targeted treatment ineffective. Although traditional bulk tumor sequencing approaches have identified essential genes and pathways that play important roles in GBM tumorigenesis, they provide limited insight into the cellular diversity and molecular complexity of tumor cells.

Recent advances in single-cell analysis methods provide an avenue to explore the underlying mechanism of normal biological processes and diseases that were not previously recognized at the population level. Single-cell RNA sequencing (scRNA-seq) generates gene expression profiles at the resolution of an individual cell [4] and has been comprehensively applied to characterize molecular features during differentiation [5,6,7], development [8,9,10], and cancers [11,12,13]. It has not only elucidated the composition of multiple cell types in normal tissues [14,15], but also dissected the complex multicellular ecosystem of cancers [16,17], which reveals many new insights into the molecular and cellular diversity of normal and tumor cells. For example, Liu and colleagues deployed single-cell RNA-seq of the human brain to provide a comprehensive annotation and quantification of long non-coding RNAs (lncRNAs) at a greater resolution [18]. In contrast to their low expression levels detected in bulk samples, lncRNAs were expressed at levels comparable with those of mRNAs in individual cells and showed the cell-type and single-cell specificity, which was confirmed by in situ hybridization. Moreover, they found one of such lncRNA, LOC646329, that could regulate cell proliferation, suggesting their important functions in human brain. Torre et al. performed single-cell analysis on a melanoma cell line to explore rare cell gene expression patterns [19]. They observed that some resistance markers such as *EGFR*, *AXL*, *WNT5A* and *NGFR*, which showed overall low average expression across all cells, had high expression levels in a subset of cells. These results indicate that genes with cell subset-specific expression patterns would have crucial effects on pivotal biological activities.

Given the important functions of these genes in specific cell subsets, the above phenomenon, which is commonly disregarded in studies, remains valuable and mysterious. Many issues urgently need to be answered. Can this phenomenon be generally observed in GBM and other cancers? What are the functions and influences of these specific genes on tumor biology?

Here, we used scRNA-seq data to comprehensively explore the expression patterns of rare genes in GBM and dissect their biological significance. We demonstrated that rare genes, which showed high levels only in subsets of cells, were widely present in multiple cancer types. Among identified rare genes, many were associated with cancer-related processes. Furthermore, rare genes present cancer stem cell-specific expression patterns, implying their potential association with cancer initiation. Further, we observed that rare genes were enriched in a subset of 17 cells which showed high cell cycle and invasive activity, indicating these rare genes may reflect or even endow the proliferation and invasive potential of GBM cells.

## 2. Results

### 2.1. Single-Cell Profiling Uncovers Rare Genes in GBM

We retrieved 576 single-cell RNA-seq profiles from five primary GBM patients and their corresponding population-level profiles [11]. After strict quality control and removal of potential non-tumor cells (see methods, Appendix A), we retained 350 tumor cells with 18906 protein-coding genes (PCGs) and 10903 lncRNAs. We observed high correlation between average expression of single cells and that of bulk samples (Figure 1A,B and Appendix A), suggesting reliability of the single-cell data, which was accordant with the results from Patel et al. However, a low correlation between single cells was observed (Appendix A), suggesting high transcriptional heterogeneity among individual cells. Through interrogating the expression distribution of each gene in single cells, we observed bimodal distributions for PCGs and long-tailed unimodal distributions for lncRNAs (upper panel in Figure 1C and Appendix A). Although most genes showed low expression levels, some of them were highly expressed in a small proportion of cells, which could not be captured by bulk profiles (lower panel in Figure 1C, Figure 1D and Appendix A). Subsequently, we investigated the distribution of average expression levels of detected PCGs and lncRNAs across cells. In contrast to the large expression difference in bulk profiles (lower panel in Figure 1C), lncRNAs and PCGs showed comparable expression levels in cells expressing them (Figure 1E and Appendix A).

Next, we attempted to investigate the distributions of proportions of cells expressing PCGs or lncRNAs. Overall, cell proportions of lncRNAs were less than those of PCGs in general (Figure 1F and Appendix A). Furthermore, we calculated an average non-zero expression for each PCG and lncRNA, and divided all the PCGs or lncRNAs into four groups on the basis of expression levels of PCGs. Interestingly, there were substantial lncRNAs and a proportion of PCGs expressed in a small population of cells, even in the high expression group (Figure 1G and Appendix A). We suspected these PCGs/lncRNAs abundant in few GBM cells, which we considered as rare genes, might play important roles in tumorigenesis. Then we extracted them from the high expression group and retained those expressed in a small proportion of cells (less than 20%). We found several critical cancer-related genes among these rare genes. For instance, *HRAS* was expressed abundantly in few cells in MGH26, MGH28 and MGH29, while activation of *HRAS* could induce the formation of GBM-like disease [20]. LncRNA *CRNDE*, highly expressed in few cells in MGH26 and MGH28, was previously found to promote glioma cell growth and invasion through mTOR signaling and might serve as a potential novel therapeutic target for glioma [21]. Furthermore, we investigated the expression patterns of cancer-related PCGs obtained from the cancer gene census (CGC) [22] and cancer-related lncRNAs from lnc2cancer database [23]. We found a proportion of cancer-related PCGs (Figure 1H and Appendix A) and cancer-related lncRNAs (Figure 1I and Appendix A) were expressed abundantly in few GBM cells. These results suggested that these rare genes were potentially biologically significant.

### 2.2. Rare PCGs/lncRNAs Present Extensively in Various Cancers

To inspect whether rare genes were widely present in human cancers, we further analyzed single-cell RNA-seq data sets of multiple cancer types (Appendix A), including breast cancer (BRCA, 563 cells, 11 samples), melanoma (307 cells, three samples), colorectal cancer (CRC, 375 cells, 11 samples) and another glioblastoma data (GBM2, 3589 cells, four samples). Through a single cell processing pipeline (Appendix A), we constructed the single-cell transcriptomes and confidently distinguished 1766 malignant cells from non-malignant cells as previously described [11,12]. Based on the expression profiles of malignant cells of each sample, we estimated the average non-zero expression and the cell proportion for each PCG/lncRNA. Strikingly, we observed the existence of substantial rare PCGs/lncRNAs across samples in all cancer types (Figure 2A,B). Notably, these rare lncRNAs, with cell proportions below 0.2 and average non-zero expression above the third quartile of that of PCGs, occupied large proportion (mean = 87.13%, range from 78.16% to 94.75%) of the abundantly expressed lncRNAs across all cancer types. Further analysis showed that a large number of rare PCGs/lncRNAs were shared by different cancer types (Figure 2C,D), with 58 rare PCGs and 115 rare lncRNAs shared by all of the five data sets. Some of them have been widely considered to be associated with the development of cancer. For instance, *SALL4* has been verified to be associated with cell migration, metastasis, invasion, drug resistance in these cancers [24]. Taken together, these results support that rare PCGs/lncRNAs, which are highly expressed in a small number of cells, were prevalent in various cancers.

### 2.3. Systematical Identification of Rare Genes in GBM

Extensive existence of rare genes in various cancers implied their potential biological significance in cancers. Thus, we focused on the GBM dataset to deeply investigate rare genes. We pooled all cells from the four patients and designed a framework to systematically identify rare genes. We first developed a classification model for each cell to filter expression noise for lncRNAs, which were potentially caused by genomic DNA contamination and incompletely processed RNA [25] (see methods). Consequently, 289 cells with the average AUC (area under curve) value larger than 0.8 were retained. We didn’t apply the classification model to PCGs because of their relative insusceptibility to sequencing noise. Instead, we removed PCGs which were detected in less than two cells with expression levels larger than one (see Methods). To systematically identify rare genes in GBM, we performed permutation tests to screen genes with significantly high average non-zero expression and low cell proportion (see methods). We defined the thresholds for average non-zero expression and cell proportion as the 99% and 1% quantiles of their null distributions, respectively (Figure 3A,B). Moreover, we removed PCGs and lncRNAs that were detected in less than 10 cells to avoid potential transcriptional noise (Figure 3C,D). Among the remaining genes, some showed high cell proportions in at least one individual (Figure 3E,F), which we suspected as potential abundant genes and here were incorrectly considered as rare genes due to the limited samples. Thus, we further removed genes with cell proportion larger than 20% in individual samples. As a result, we finally identified 98 rare genes, including 51 rare PCGs and 47 rare lncRNAs (Appendix A). Interestingly, a moderate proportion of rare PCGs (27.5%) and rare lncRNAs (19.1%) were differentially expressed in GBM tissue samples from TCGA (see methods, Figure 3G–I). Notably, differentially expressed PCGs and lncRNAs were significantly enriched in rare PCGs and rare lncRNAs (hypergeometric test, *p* = 1.1 × 10^−3^ for PCGs and *p* = 6.4 × 10^−4^ for lncRNAs), respectively. Moreover, among these rare genes, the overall survival (OS) of patients with high expression levels of *CYB5R2*, *TPPP3* or *TSSC4* were significantly shorter than those with low expression levels of corresponding rare genes *(p* < 0.0001, *p* = 0.00036 and *p* = 0.03, respectively, Figure 4A). Especially, patients with high expression levels of *CYB5R2*, *TPPP3* had worse disease-free survival (DFS) than those with low expression levels (*p* = 0.013 and *p* = 0.039, respectively, Figure 4B). These results suggested potential implication of rare genes in GBM tumorigenesis, progression, and prognosis, which need to be further investigated.

### 2.4. Rare Genes Are Related with Cancer Stemness of GBM

Since cancer stem cells (CSCs), which have the ability to self-renew, are a special and rare subpopulation of tumor cells [26], we investigated whether rare genes had specific expression pattern in cancer stem-like cells of GBM. To this end, we first obtained 36 marker sets of human stem cells from StemChecker [27] and then selected a 51-gene stem cell signature, which was significantly differentially expressed between all three pairs of cultured gliomasphere cells (GSC) and differentiated cells (DGC) models (Figure 5A). We could also distinguish CSCs and most of the tumor cells using these markers, supporting the reliability and efficiency of this signature. Notably, we found a fraction of tumor cells mixed in the CSC cluster (Figure 5B), implying the existence of GBM cancer stem-like cells. Similar results were observed from principal component analysis (PCA) using this signature (Appendix A), in which the second component (PC2) could well separate two types of cells. Moreover, PC2-related genes were significantly enriched in stemness functions, such as cell adhesion, cell proliferation and neuron projection regeneration (Appendix A). Finally, we calculated the stemness scores by subtracting the average expression of all genes from that of the 51 markers, which were distinguished from that calculated based on 1000 random marker sets (Appendix A). The above results indicated that the 51 markers could be used to identify potential cancer stem-like cells in GBM.

Therefore, we performed hierarchical clustering using this signature to identify 36 cancer stem-like cells from 289 tumor cells across MGH26, MGH28, MGH29 and MGH30. In MGH28, there were 14 cancer stem-like cells and 64 tumor cells. We found that 2 rare PCGs expressed only in cancer stem-like cells of MGH28 and similar results were obtained for other three samples (Figure 5C, Appendix A). Among these rare genes, the membrane protein SLCO4A1 has been reported to be overexpressed in pluripotent stem cells but suppressed in differentiated cells [28], which was coincident with our findings (Figure 5D). These results indicated the potential association of rare genes with CSCs.

### 2.5. Rare Genes Reveal a Subpopulation of Cancer Cell with High Proliferation Activity and Invasive Potential

To examine the expression changes of rare genes during GBM progression, we utilized Monocle2 [6] to perform pseudotemporal ordering of single cells and construct cancer progression trajectory with a tree-like structure (Appendix A), resulting in seven distinct cell “states” (State1-State7). We found a part of rare genes mainly expressed in a subset of 17 cells at the end of the trajectory, which accounted for the majority of State7 cells (Figure 6A). Most of these rare genes were associated with cell cycle, such as *TK1*, *CDCA5* and *CDC45* (Figure 6A and Appendix A), implying these cells may have high cell cycle activity. To precisely characterize cycling cells, we calculated the G1/S and G2/M scores for each cell using gene signatures [16] that denote G1/S or G2/M phases. Cell cycle phase-specific signatures were highly expressed in this cell subset (Figure 6B), which was consistent with the state-specific genes and their enriched functions (Appendix A).

To further determine the identity of this cell subset, we used SCDE [29] to identify the differentially expressed genes (DEGs). Besides the cell division-associated genes like *CCNB2* and *BUB1B*, we also found genes such as *TPX2* [30], *BIRC5* [31], *TOP2A* [32], *SPAG5* [33] and *HMGB1* [34], were among the top 10 highly expressed genes in the cell subset (Appendix A, Appendix A), which were reported to be directly or indirectly associated with cancer invasion. Therefore, we assumed that these cells also had high invasive potential. To validate this assumption, we used invasion-associated signatures obtained from the molecular signatures database (MSigDB) [35,36] to calculate the invasive potential score (IPS, see Methods) and performed cluster analysis (Figure 6C). The fourth cluster (C4) contained all the 17 cells and showed the highest expression of invasion-associated signatures. Moreover, we found *ZEB1* and *HNRNPC*, both of which were experimentally verified to be involved in invasion and migration of GBM cells [37,38], were significantly overexpressed in C4 (Appendix A), suggesting that this cell subset also had high invasive potential.

## 3. Discussion

Previous studies have focused on revealing driver genes and aberrant regulatory programs [39] that were commonly or frequently observed in the majority of samples. However, less attention has been paid to the rare genes because of their relatively low detection rate. Notably, despite the common disregard of rare genes, some studies [18,19] have implied their important functions in tumor development and heterogeneity. Here, our study revealed the general existence of rare genes in GBM and three other cancers utilizing scRNA-seq data. Further, we designed a strategy to systemically identify rare genes in GBM. Importantly, our results demonstrated the potential relation of rare genes with cancer stemness and metastatic activity.

We found several rare genes were specifically expressed in cancer stem cells of GBM, some of which are associated with CSC properties. Moreover, one key step for functional studies on CSCs is to find specific markers to distinguish them from tumor cells. Single-cell analysis enabled us to identify rare genes that could be potential markers for CSCs. In future research, we will extend our studies to explore the specific expression pattern of rare genes in CSCs of other cancer types.

In the traditional models of tumorigenesis, invasion or metastasis is a late event during tumor evolution. However, some studies have shown that epithelial-mesenchymal transition (EMT) and dissemination can occur before metastasis detected by histological examination. Robert Weinberg and his colleagues considered that selection pressure within the primary tumor environment would not contribute to the development of a metastatic phenotype. Therefore, they proposed an alternative model that tumor cells can acquire the ability to metastasize at a much earlier than previously recognized stage in the tumor evolution [40,41]. Moreover, Andrew D. Rhim et al. used a Cre-lox-based mouse model of PDAC to study the metastatic mechanism. They found that tagged cells traversed the basement membrane and entered the bloodstream before invasive behavior was detectable by standard histology [42], suggesting that tumor cells could acquire invasive potential in primary tumors. Similar results were observed in breast cancer [43] and melanoma [44]. In our study, the expression profile of rare genes helped us to identify a subset of tumor cells that had high invasive potential scores. Notably, based on the pseudotime ordering, this cell subset was located in an individual branch of the trajectory. These results indicated that some special tumor cells have already obtained the ability to invade and metastasize in GBM primary tumor, although the invasive and metastatic behavior have not been observed, supporting the alternative model. However, little is known about the molecular characterizations of this cell subset and the underlying mechanism of their generation. 

The limitation of our study is the small number of tumor cells. In the future, we will further integrate more single-cell RNA-seq data sets to explore the functional significance of rare genes in cancers and expand our study into more cancer types to comprehensively dissect the contribution of rare genes to the tumor evolution.

## 4. Materials and Methods

### 4.1. Processing of Publicly Available Single-Cell RNA-Seq Data of GBM

We retrieved raw single-cell RNA sequencing data of GBM from SRA accession SRP042161 [11], including 576 cells from five primary GBM patients (MGH26, MGH28, MGH29, MGH30 and MGH31) and their corresponding bulk samples, as well as 192 single gliomasphere cells from two GBM cancer stem cell lines and 3 pairs of bulk samples from cultured gliomasphere (GSC) and differentiated (DGC) samples. To retrieve the transcriptomic profiles, we first built a reference transcriptome based on the GENCODE v19 annotation [45] using RSEM module rsem-prepare-reference (version 1.2.28) [46]. Paired-end 25bp reads were then mapped to the reference transcriptome by Bowtie (version 1.1.1, with parameters -n 0 -e 99999999 -l 25 -I 1 -X 2000 -a -m 15 -S) [47], and expression quantification of all genes from Gencode v19 were performed using RSEM module rsem-calculate-expression (version 1.2.28, using the option estimate-rspd and default parameters) [46]. Transcripts per million (TPM) values of PCGs and lncRNAs were extracted based on gene biotypes *protein_coding* and *processed_transcript, lincRNA, 3prime_overlapping_ncrna, antisense, non_coding, sense_intronic* or *sense_overlapping*, respectively. Log2 transformed TPM values with an offset of 1 were used to denote expression levels. We excluded low-quality cells with less than 200,000 aligned reads or with less than 3000 detected genes (log2(TPM + 1) > 0). As a result, we retained 411 GBM cells and 129 CSCs with 19,672 PCGs and 12,623 lncRNAs.

### 4.2. Removing Potential Non-Tumor Cells

To distinguish non-tumor cells from malignant tumor cells, we inferred copy number variations for each cell by averaging expression levels of genomically adjacent genes in fixed bins [11]. In brief, all genes were sorted according to their genomic coordinates, then a relative copy number for each gene was calculated by averaging expression levels of its 100 genomically flanking genes. Thus, we estimated a copy number variation (CNV) vector for each cell, and then we centered these CNV vectors separately. We also performed the same analysis for normal brain samples from the GTEx portal [48]. However, we inferred CNV for one brain reference sample by averaging expression levels across all brain samples instead of inferring CNV for each sample. Taking this brain reference CNV as a control, we then employed hierarchical clustering to distinguish 11 non-tumor cells from malignant tumor cells. Thus, we removed these 11 potential non-tumor cells and retained 400 malignant tumor cells, including 106 cells from MGH26, 96 cells from MGH28, 75 cells from MGH29, 73 cells from MGH30 and 50 cells from MGH31.

### 4.3. Filtering Expression Noise for lncRNAs Based on a Classification Model

We developed a classification model to filter expression noise, which could be caused by genomic DNA contamination and incompletely processed RNA, for lncRNAs inspired by Iyer et al. [25]. First, we extracted lncRNAs, which were detected in more than 80% cells, as the positive set, and lncRNAs which were detected in no more than 2 cells as the negative set. Then we established a model for each cell based on two dimensions, namely relative abundance and exon coverage. Relative abundance was calculated for all lncRNAs detected in the cell according to the empirical distribution of their expression. Exon coverage were also calculated for these lncRNAs as the ratio of the number of exons covered by reads to the number of all exons. Based on the abundance-coverage coordinate system, we computed bivariate kernel density estimates for the positive set and the negative set, separately. Then, we imputed densities for other lncRNAs using a linear interpolation method based on density distribution in the positive set and the negative set, respectively. Next, likelihood ratios were calculated by dividing density estimates in positive set by density estimates in negative set after adding a small value (1 × 10^−10^) to avoid floating-point overflow errors. We defined a cutoff for the likelihood ratios in each cell to classify lncRNAs into true expression set and noise expression set. The likelihood ratio with max balanced accuracy (average of sensitivity and specificity) of the classifier performance was selected, and lncRNAs with likelihood below this cutoff were labeled “noise” and others “expressed”. To estimate the performance of the classifier, we adopted the five-fold cross-validation method. Finally, we retained 289 cells with average AUC (area under the curve) of testing sets larger than 0.8.

### 4.4. Identifying Rare Genes in GBM

In order to systematically identify rare PCGs and rare lncRNAs in GBM, we attempted to screen those genes whose average non-zero expression were significantly greater than expected by chance and cell proportions were significantly lower than expected by chance. Specifically, we randomly shuffled the combined expression profile of PCGs and lncRNAs for 1000 times, and then estimated null distributions for average non-zero expression levels and cell proportions through calculating these values in the 1000 scrambled profiles. We determined the threshold of average non-zero expression levels as the 99th percentile value of its null distribution and the threshold of cell proportion as the first percentile value of its null distribution. We first extracted PCGs and lncRNAs with higher average non-zero expression levels than its threshold and lower cell proportions than its threshold. Among them, we further filtered out genes detected in less than 10 cells. Finally, we obtained rare PCGs and rare lncRNAs by removing genes with high cell proportions (20%) in at least one of the four individuals (MGH26, MGH28, MGH29 and MGH30). 

### 4.5. Differential Analysis and Survival Analysis on GBM Tissue Samples from TCGA

Read count matrix of 170 GBM samples and 5 control samples from TCGA were downloaded from https://osf.io/gqrz9/ [49]. Differential analysis was performed using DESeq2 method [50]. Genes with fold change >4 and FDR adjusted *p*-value < 0.05 were determined as differential genes.

The GBM data for survival analysis was obtained from the public cBio Cancer Genomics Portal (http://www.cbiop ortal.org) [28,51]. The overall survival and disease-free survival were used as the end points. The Kaplan-Meier method was used for the visualization purposes and the differences between survival curves were calculated by log-rank test. *P* values less than 0.05 were considered to be statistically significant. All of these statistical analyses were performed using R software (http://www.rproject.org), version 3.4.4.

### 4.6. Single Cell Trajectory Analysis

Monocle2 (version 2.6.3) algorithm [6] was used to construct single cell pseudotime trajectory using 2500 most highly variable genes with the default parameters. Monocle2 uses Reversed Graph Embedding, a machine learning technique to learn a parsimonious principal graph, reduces the given high dimensional expression profiles to a low-dimensional space. Single cells are projected onto this space and ordered into a trajectory with branch points. As in Monocle2, cells in the same segment of the trajectory have the same “state”. This resulted in seven distinct cell states. We performed differential expression analysis between each state vs. the remaining six using the differentialGeneTest function to identify state-specific genes. These genes were divided into four clusters using plot_pseudotime_heatmap function with the default parameters. We used the hypergeometric test to identify cluster-related functional gene sets in the Molecular Signatures Database (MSigDB) [35,36], which showed significant overlap with genes in each cluster.

### 4.7. Cell Cycle Analysis

Gene sets reflecting the G1/S and G2/M phases of the cell cycle were obtained from Tirosh et al. [16]. Genes with no expression across all cells were filtered out. The average expression of each gene set was calculated as G1/S and G2/M scores, respectively, shown in Figure 6.

### 4.8. Single Cell Differential Expression Analysis

We used the SCDE software package (version 2.2.0) [29] to identify the significantly highly expressed genes in the cell subgroup. SCDE models the counts for each gene using a mixture of a Negative Binomial distribution and a Poisson distribution. It uses a Bayesian approach to compute the posterior probability that a gene shows differential expression between two conditions. Since the SCDE algorithm requires integer values that should not be normalized, the raw read counts were provided as input. Genes with zero reads in more than half of the compared cells were discarded. We considered the genes with absolute cZ more than 1.96 as the significantly differentially expressed genes.

### 4.9. Invasive Potential Score (IPS)

We obtained four sets of metastatic signatures from the MSigDB. Genes with no expression across all cells were filtered out. We calculated four invasive potential scores as the average expression of the four gene sets.

## 5. Conclusions

In summary, our results reveal the prevalence of rare genes and their functional implications in cancers, provide a more in-depth understanding of the underlying biology of rare genes in GBM, and new strategies for the diagnosis and treatment of GBM. 

## Figures and Tables

**Figure 1 cancers-11-02025-f001:**
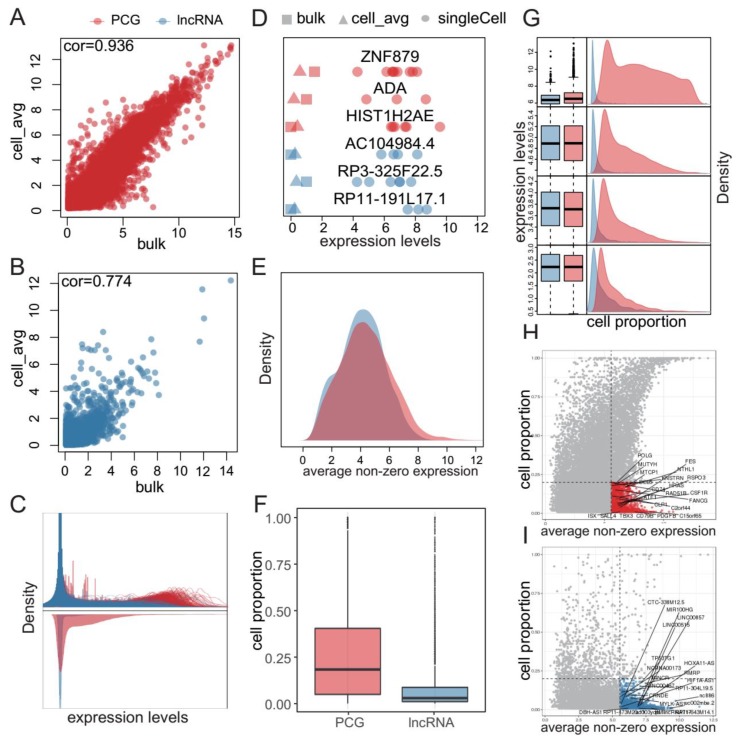
Comprehensive characterization of transcriptional patterns of protein-coding genes (PCGs) and lncRNAs in GBM single cells. (**A,B**) Correlation between average expression of all single cells and that of bulk samples based on PCGs (**A**) and lncRNAs (**B**). Each point represents a PCG (red) or lncRNA (blue). (**C**) Distribution of expression levels of PCGs (light red lines) and lncRNAs (light blue lines) in single cells (upper panel) and the bulk sample (lower panel) from MGH26. (**D**) Examples of PCGs and lncRNAs which were abundant in single cells while rarely detected in bulk samples from MGH26. (**E**) Distribution of average non-zero expression levels of PCGs and lncRNAs in MGH26. (**F**) Distribution of cell proportion of PCGs and lncRNAs among cells from MGH26. (**G**) Distribution of cell proportion for PCGs and lncRNAs grouped by average non-zero expression quartiles of PCGs among cells from MGH26; Expression levels of the four groups increased from bottom to top. (**H–I**) Several cancer-related PCGs (**H**) and lncRNAs (**I**) showed abundant expression in few cells among cells from MGH26.

**Figure 2 cancers-11-02025-f002:**
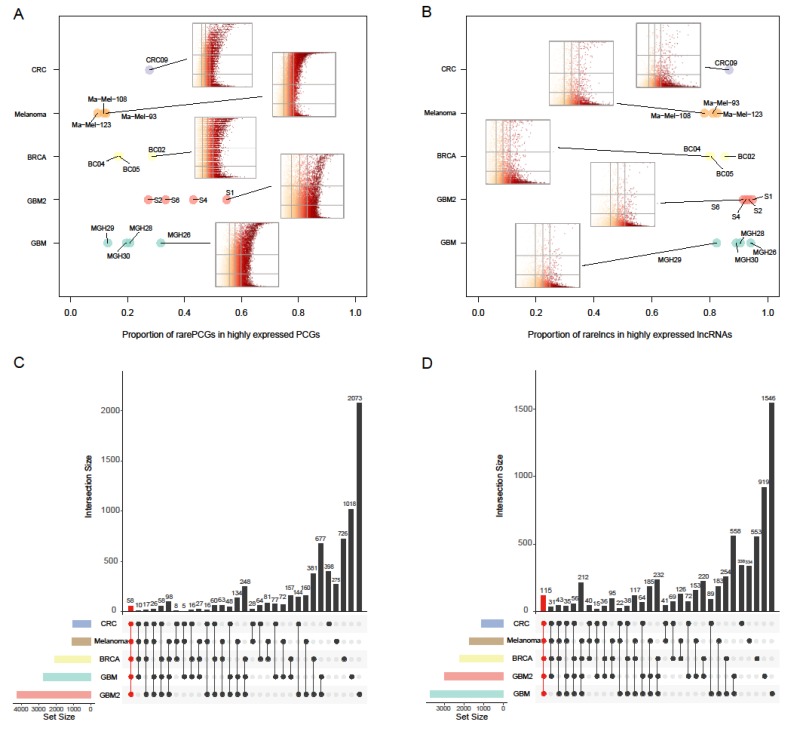
Rare protein-coding genes (PCGs)/lncRNAs widely present in GBM, BRCA, Melanoma and CRC. (**A,B**) Rare PCGs (**A**) or rare lncRNAs (**B**) account for substantial abundantly expressed PCGs or lncRNAs in each sample, respectively. Each larger point represents one sample, while each color represents one cancer type. The horizontal axis represents the proportion of rare genes in all highly expressed ones for each sample. The embedded scatterplots show the cell proportions (vertical axis) and mean non-zero expression levels (horizontal axis) of each PCG (**A**) and lncRNA (**B**) in designated samples. The vertical lines represent the quartiles of PCG expression as 0.25, 0.5 and 0.75, while the horizontal lines represent 0.2 and 0.5. The color represents the mean non-zero expression levels of genes, where yellow present low expression levels and red present high expression levels. (**C,D**) The upper barplot showing the overlaps of rare PCGs (**C**) and rare lncRNAs (**D**) between different cancer types. The number of shared cancer types was shown by the number of points in the below panel. For example, the red bar in (**C**) means there were 58 rare PCGs shared by five cancer types. The color bars in the lower-left panel represent the numbers of all rare PCGs (**C**) and rare lncRNAs (**D**) identified in at least one sample for each cancer type.

**Figure 3 cancers-11-02025-f003:**
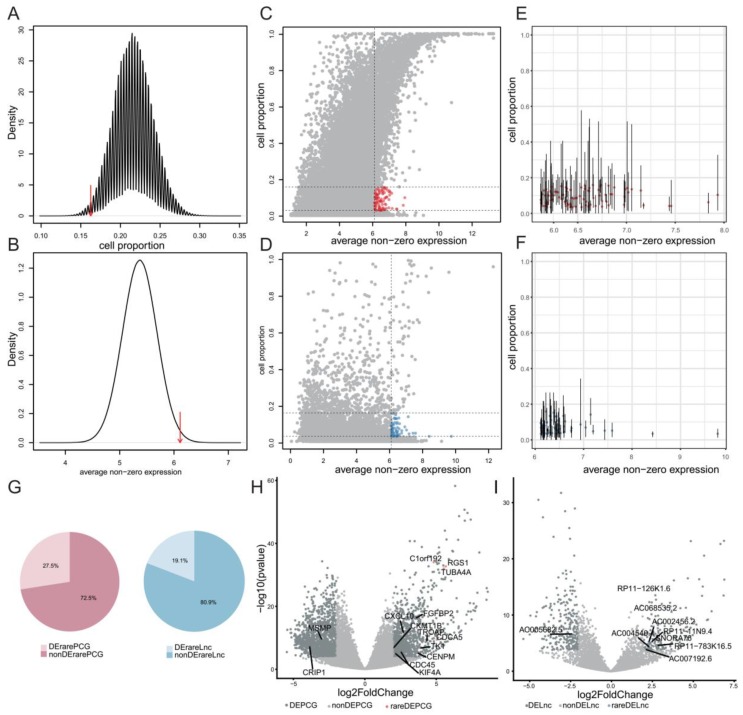
Screening rare genes in GBM. (**A**) Null distribution of average non-zero expression. Red arrow indicates the threshold of high average non-zero expression level as the 99th percentile value; (**B**) Null distribution of cell proportion. Red arrow indicates the threshold of low cell proportion as the 1th percentile value; (**C,D**) Scatter plots showing the distribution of cell proportion against average non-zero expression levels of protein-coding genes (PCGs) (**C**) and lncRNAs (**D**), in which red and blue points represent rare PCGs and rare lncRNAs, respectively; (**E,F**) Some of the rare PCGs (**E**) and rare lncRNAs (**F**) identified in all cells show high cell proportions. Points denote cell proportions and average non-zero expression levels across all cells, and vertical lines denote ranges of cell proportions in the four individuals; (**G**) Pie plots showing the proportion of rare PCGs and rare lncRNAs which were differentially expressed in GBM tissue samples from TCGA. DErarePCG, differentially expressed PCG. DErareLnc, differentially expressed lncRNA; (**H,I**) Volcano plots representing differentially expressed PCGs (**H**) and lncRNAs (**I**) in GBM tissue samples from TCGA. Grey points denote non-differentially expressed genes, light steel blue points denote differentially expressed genes, red points denote differentially expressed rare PCGs and sky blue points denote differentially expressed rare lncRNAs.

**Figure 4 cancers-11-02025-f004:**
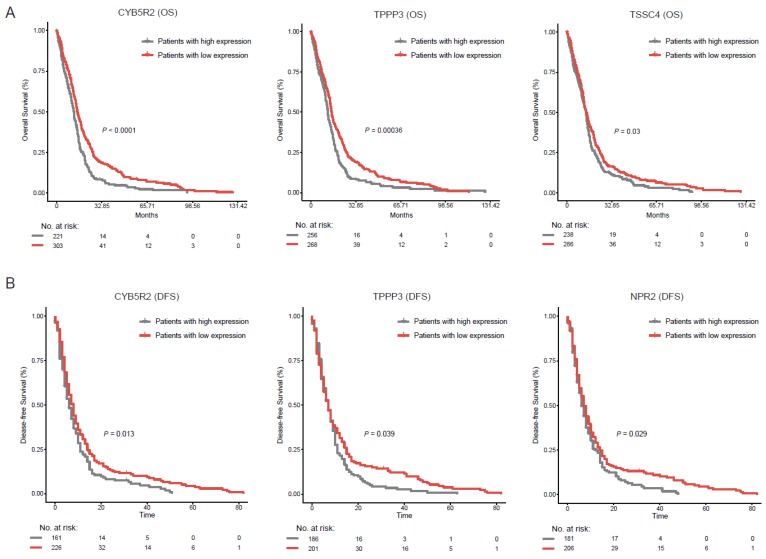
The associations of rare genes with clinical outcome. (**A**) Comparison of overall survival among patients with high expression levels of rare genes (gray line) and those with low expression levels of corresponding rare genes (gray line) by Kaplan–Meier analysis (with log-rank P values) in the cohort of GBM patients from TCGA. The rare genes include *CYB5R2*, *TPPP3* and *TSSC4*. The patients were divided into two groups based on the average expression level of the corresponding rare gene across all patients. (**B**) Comparison of disease-free survival among patients with high expression levels of rare genes (gray line) and those with low expression levels of corresponding rare genes (gray line) by Kaplan–Meier analysis (with log-rank P values) in the cohort of GBM patients from TCGA. The rare genes include *CYB5R2*, *TPPP3* and *NPR2*. The patients were divided into two groups based on the average expression level of the corresponding rare gene across all patients.

**Figure 5 cancers-11-02025-f005:**
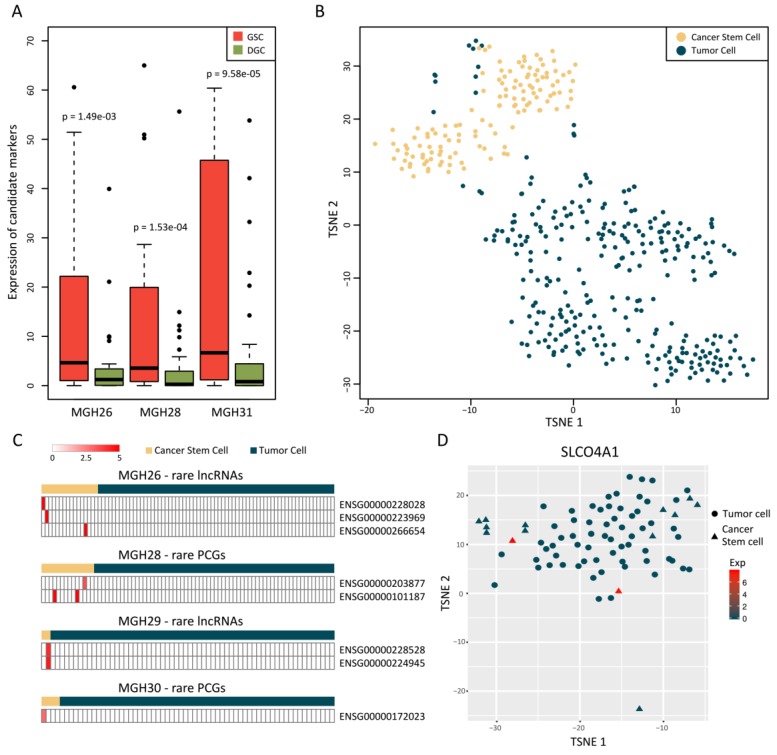
Rare genes specifically expressed in cancer stem cells. (**A**) The 51-gene stem cell signature was significantly differentially expressed between three pairs of gliomasphere cells (GSC) and differentiated cells (DGC) culture models; (**B**) T-SNE plot based on the 51-gene signature showed the separation of cancer stem cells (CSCs) and most of the tumor cells; (**C**) Heatmaps displaying rare genes specifically expressed in cancer stem-like cells of GBM. Each row represents a rare gene and each column represents a cell of the corresponding sample. The color represents the expression levels of rare genes, where white means low expression level and red means high one; (**D**) The T-SNE plot showing examples of a rare protein-coding genes (PCG), *SLCO4A1*, which was exclusively expressed in cancer stem-like cells of MGH28. The circle represents tumor cells and the triangle represents identified cancer stem-like cells. The color represents the expression level of *SLCO4A1* in each cell, where dark blue means low expression level and red means high one.

**Figure 6 cancers-11-02025-f006:**
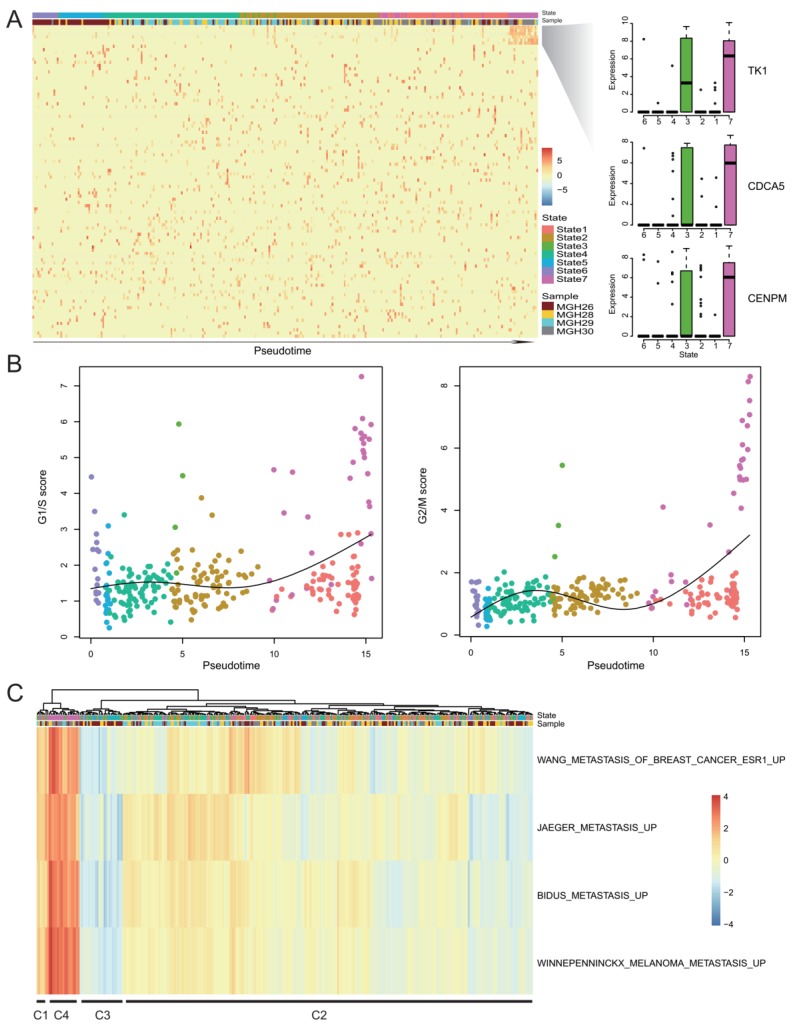
Identification of a subset of cells with high cell cycle and invasive activities. (**A**) Heatmap shows rare gene expression dynamics during GBM progression. Rare genes (row) are clustered and cells (column) are ordered according to the pseudotime. The boxplots in the right panel show the expression levels of three rare genes in cells with each state. (**B**) Scatterplots show the G1/S (left) and G2/M (right) scores of cells which are ordered according to the pseudotime. Each point was colored by states, which was the same as (**A**). (**C**) Heatmap shows invasive potential score of cells which are clustered into four groups (C1-C4).

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
