# Peer review of "Discovering Rare Genes Contributing to Cancer Stemness and Invasive Potential by GBM Single-Cell Transcriptional Analysis"

_cancers, 2019, doi:10.3390/cancers11122025_

Round 1
Reviewer 1 Report
The authors have responded well to my concerns. There are still a few minor grammatical mistakes.
Author Response
Thank you very much for your comments. According to your suggestion, we have our manuscript checked by a native English speaker and try our best to correct grammatical mistakes in the revised manuscript.
Reviewer 2 Report
The authors have adequately addressed all previously raised issues. The data on survival is quite strong and increases my enthusiasm. While functional studies would be certainly out of the scope of this manuscript, the authors are encouraged to highlight these data in the abstract and include them as a main figure (instead of as supplemental fig. 4).
Reviewer 3 Report
The authors have carefully adressed the comments raised during the 1st round of review. There is a need for English Language proof reading.
Author Response

(The authors gave the same response as above.)
